# A2Perf: Real-World Autonomous Agents Benchmark

## Abstract

Autonomous agents and systems cover a number of application areas, from robotics and digital assistants to combinatorial optimization, all sharing common, unresolved research challenges. It is not sufficient for agents to merely solve a given task; they must generalize to out-of-distribution tasks, perform reliably, and use hardware resources efficiently during training and on-device deployment, among other requirements. Several major classes of methods, such as reinforcement learning and imitation learning, are commonly used to tackle these problems, each with different trade-offs. However, there is currently no benchmarking suite that defines the environments, datasets, and metrics which can be used to develop reference implementations and seed leaderboards with baselines, providing a meaningful way for the community to compare progress. We introduce A2Perf—a benchmarking suite including three environments that closely resemble real-world domains: computer chip floorplanning, web navigation, and quadruped locomotion. A2Perf provides metrics that track task performance, generalization, system resource efficiency, and reliability, which are all critical to real-world applications. In addition, we propose a data cost metric to account for the cost incurred acquiring offline data for imitation learning, reinforcement learning, and hybrid algorithms, which allows us to better compare these approaches. A2Perf also contains baseline implementations of standard algorithms, enabling apples-to-apples comparisons across methods and facilitating progress in real-world autonomy. As an open-source and extendable benchmark, A2Perf is designed to remain accessible, documented, up-to-date, and useful to the research community over the long term.

## 1 Introduction

Autonomous agents observe their environment, make decisions, and perform tasks with minimal human interference [57]. These agents have been successfully evaluated across a wide range of application domains. However, developing algorithms for autonomous agents that can be deployed in real-world scenarios presents significant challenges [14]. These challenges include dealing with high-dimensional state and action spaces, partial observability, non-stationarity, sparse rewards, and the need for safety constraints. Furthermore, real-world environments often have multiple objectives, require sample efficiency, and necessitate robust and explainable decision-making. Addressing these challenges is crucial for productionizing reinforcement learning algorithms to real-world problems.

To enable researchers to develop algorithms with real-world deployment considerations in mind, there is a need for benchmarks that incorporate practical metrics. These include metrics such as the compute required for training and inference, wall-clock time, and effort expended on data collection. While there are existing benchmarks for autonomous agents [25; 65; 34; 4; 9; 58], most only evaluate an agent's raw performance on the same task on which it was trained, without considering numerous other metrics that matter in real-world production training and deployment scenarios.

In this paper, we introduce A2Perf[1], a benchmarking framework that aims to bridge the gap between algorithms research and real-world applications by providing a comprehensive evaluation platform for autonomous agents, thereby expanding the applicability of reinforcement learning to a wide range of practical domains. In addition, it comes equipped with a critical set of metrics for fair assessment.

---

[1]A2Perf code: `https://anonymous.4open.science/r/A2Perf-2BFC`

| Benchmark | Metrics | | | | Realistic Tasks | Offline Datasets |
|---|---|---|---|---|---|---|
| | Generalization | System | Data Cost | Reliability | | |
| A2Perf | ✓ | ✓ | ✓ | ✓ | ✓ | ✓ |
| D4RL [16] | ✓ | ✗ | ✗ | ✗ | ✓ | ✓ |
| Meta-World [65] | ✓ | ✗ | ✗ | ✗ | ✓ | ✗ |
| CoinRun [10] | ✓ | ✗ | ✗ | ✗ | ✗ | ✗ |
| DM Control [58] | ✗ | ✗ | ✗ | ✗ | ✗ | ✗ |
| Safety Gym [32] | ✗ | ✗ | ✗ | ✗ | ✓ | ✗ |
| ALE [4] | ✗ | ✗ | ✗ | ✗ | ✗ | ✗ |
| MineRL [25] | ✓ | ✗ | ✗ | ✗ | ✗ | ✓ |
| OpenAI Gym [5] | ✗ | ✗ | ✗ | ✗ | ✗ | ✗ |
| Loon Benchmark [18] | ✓ | ✗ | ✗ | ✗ | ✓ | ✓ |

**Table 1:** A2Perf compared to existing benchmarks that evaluate autonomous agents. Checkmarks (✓) indicate the presence of a feature or metric, while crosses (✗) denote its absence. A2Perf distinguishes itself by including metrics for generalization, system resource efficiency, data cost, and reliability, in addition to providing realistic tasks and offline datasets. The selected domains in A2Perf are designed to closely mirror real-world challenges, ensuring the relevance and transferability of the benchmark results to practical applications.

A2Perf incorporates three challenging domains based on prior work [13; 43; 23] that closely mirror scenarios that have been demonstrated in the real world: computer chip-floorplanning, website form-filling and navigation, and quadruped locomotion. In addition, these domains were chosen because they inherently exhibit a small Sim2Real gap. The computer chip-floorplanning domain [42; 43] was used to help create an iteration of Google's tensor processing unit[2], where the autonomous agent optimizes the layout of chip components. In the website form-filling and navigation domain [22; 23], agents autonomously navigate and interact with websites in a Google Chrome[3] browser, making it identical to real-world web navigation. The quadruped locomotion domain [49] has demonstrated successful transfer of learned walking gaits to the Unitree Laikago[4] robot.

Furthermore, to address the metrics gap, A2Perf provides an open-source benchmarking suite that evaluates agents across four key metric categories: (1) data cost, which quantifies the effort required to gather training data for imitation learning, (2) application performance, relating to the quality of the agent's task-specific execution, and it's ability to generalize to tasks that it was not explicitly trained to perform; (3) system resource efficiency, focusing on the hardware resources used during training and inference; and (4) reliability, denoting the consistency of an agent's performance over training and inference. While three domains and for classes of metrics are currently available, A2Perf allows for straightforward expansion to benchmark on custom domains and for custom metrics.

Our experimental evaluation yields valuable insights into the real-world applicability of autonomous agents across diverse domains. In the web navigation domain, we explore the feasibility of deploying agents by analyzing their inference time, power usage, and memory consumption, demonstrating that trained agents can operate with latencies comparable to human reaction times on consumer-grade hardware. Furthermore, our reliability metrics prove crucial in selecting agents for chip floorplanning and quadruped locomotion tasks. For chip floorplanning, we find that the PPO [53] algorithm provides more consistent initial placements compared to DDQN [61], reducing variability for designers. In quadruped locomotion, PPO exhibits superior stability during training, while SAC [26] demonstrates more consistent gaits during deployment, highlighting the importance of considering reliability in real-world scenarios. These findings underscore A2Perf's ability to provide a comprehensive evaluation of autonomous agents, facilitating their successful deployment in practical applications.

## 2 RELATED WORK

**Benchmarking Autonomous Agents** Table 1 offers a comparison between A2Perf and existing benchmarks, highlighting the unique contributions of our proposed benchmarking suite. Existing benchmarks for autonomous agents, such as those introduced by Brockman et al. [5]; Bellemare

---

[2]History of the Tensor Processing Unit: `https://shorturl.at/Bo71S`

[3]Google Chrome Browser: `https://www.google.com/chrome/`

[4]Unitree Laikago: `https://shorturl.at/FD6uP`

et al. [4]; Tassa et al. [58], provide diverse environments for testing various algorithms. However, these benchmarks often focus on specific types of learning algorithms or on evaluating particular desirable qualities in autonomous agents. For example, Fu et al. [16] and Gulcehre et al. [21] evaluate offline reinforcement learning [38], while Yu et al. [65] focuses on meta-reinforcement learning [62]. Similarly, Ye et al. [64] tests sample efficiency, Guss et al. [25] challenges agents on long-horizon tasks, and Cobbe et al. [10] evaluates generalization ability. While these benchmarks provide insights, they do not fully capture the challenges faced by autonomous agents in real-world applications [14]. Environments, benchmarks, and datasets have been made to foster the development of autonomous agents in real-world scenarios, such as aerial balloon navigation [18], autonomous driving [56], website navigation [23], and furniture assembly [37]. Yet, these initiatives are often domain-specific and lack the comprehensive scope needed to evaluate agents across a wide range of real-world challenges as outlined by prior work [14], which forms the basis for our work. Consequently, there remains a need for a benchmarking suite that encompasses a diverse set of tasks and environments, reflecting the complexity and variety of problems encountered in real-world applications.

**Benchmarking System Performance**   In addition to evaluating task-specific performance metrics, analyzing the end-to-end performance cost and examining the hardware resources required to apply learning algorithms on specific environments has gained significant attention [63; 47]. Benchmarks such as MLPerf [50] and DAWNBench [12] have been developed to assess various aspects of commercial deep learning workloads across training and inference, considering a diverse class of systems. Furthermore, recent studies have investigated the environmental impact of deep learning by quantifying the carbon footprint associated with training and inference using large neural network models [48]. This line of research has also extended to autonomous agents, with works like QuaRL demonstrating reduced energy consumption and emissions through lower-precision distributed training [36]. Despite these efforts, there remains a need for evaluating the system performance and energy consumption of autonomous agents to provide valuable insights into their practical feasibility and sustainability.

**Reliability Metrics for Reinforcement Learning**   Reliability is a concern in reinforcement learning (RL), as current metrics often rely on point estimates of aggregate performance, which fail to capture the true performance of algorithms and make it challenging to draw conclusions about the state-of-the-art [1; 27; 11]. The increasing complexity of benchmarking tasks has made it infeasible to run hundreds of training runs, necessitating the development of tools to evaluate reliability based on a limited number of runs [1]. For real-world deployments, reliability is essential to ensure that RL algorithms perform consistently and robustly across different conditions and environments. To assess reliability, it is essential to consider metrics across three axes of variability: time (within a training run), runs (across random seeds), and rollouts of a fixed policy [8]. By incorporating reliability metrics into A2Perf, we will be able to better assess the robustness and consistency of RL algorithms.

## 3   Evaluation Metrics

To assess autonomous agents for real-world applications, A2Perf offers a comprehensive set of metrics across four categories: data cost, application performance, system performance, and reliability. Table 2 summarizes the metrics corresponding to each category. The relative importance of these categories varies depending on the specific application domain, so in Section 4, we state which metric categories are most critical for each domain included in the initial release of A2Perf to help guide practitioners in selecting the most suitable agent for their use case.

### 3.1   Data Cost

Autonomous agents can be trained either with or without expert demonstrations. Methods that leverage expert demonstrations, such as imitation learning (IL) [29; 31; 6; 3; 33; 54], aim to learn from pre-collected datasets of human or expert agent trajectories. On the other hand, methods like online [44] and offline RL [38; 60; 45; 2] do not necessarily require expert demonstrations and instead learn through interaction with the environment or sub-optimal demonstration data.

Comparing agent performance trained using different approaches is challenging but important to gain a holistic picture of the costs and trade-offs involved. IL methods may be more sample efficient than RL methods, as they do not need to interact with the environment online. However, this perspective overlooks the *effort* required to collect expert demonstration data used for IL.

| | Data Cost | System | Reliability | Application |
|---|---|---|---|---|
| **Training** | Training Sample Cost | Energy
Power
RAM Usage
Wall-Clock Time | Dispersion (Runs)
Dispersion (Time)
Long-Term Risk (Time)
Risk (Runs)
Short-Term Risk (Time) | Episodic Returns
Generalization Returns |
| **Inference** | N/A | Inference Time
Power
RAM Usage | Dispersion (Rollouts)
Risk (Rollouts) | N/A |

**Table 2:** A2Perf assesses four categories—data cost, system performance, reliability, and application perfor­mance—during training and inference. These metrics provide a comprehensive evaluation of autonomous agents. See Section 3 for detailed descriptions of the metric categories.

To facilitate fair comparisons between these approaches, we propose the **training sample cost** metric, which quantifies the effort required to obtain offline datasets used by the agent. In this context, we denote the training sample cost of an offline dataset $D$ as $C_D$. An agent that uses samples from datasets $D_1, D_2, \ldots, D_K$ will incur a total training sample cost of `Training Sample Cost =` $\sum_{i=1}^{K} C_{D_i}$. The datasets $D_i$ could be of different *expertise* levels, meaning they contain demonstra­tions from agents or humans with varying levels of task proficiency.

The training sample cost can be measured with any metric that meaningfully represents the effort required to generate samples for imitation learning. For example, the cost could be expressed in terms of money spent on human labor or computational resources, hours invested in collecting the data, or any other relevant metric. The choice of metric may depend on the specific application and the type of data being collected since training samples can originate from a variety of sources, such as human operators [41], pre-existing policies [28], or logged experiences from different agents [17; 35].

In A2Perf, we restrict our use of the training sample cost metric to datasets generated solely from RL policies. Specifically, we define the training sample cost, $C_D$, of a dataset $D$ as the average energy consumed to train the policies that are used to generate the dataset $D$. This can be expressed as:

$$C_D = \frac{1}{|\Pi_D|} \sum_{\pi \in \Pi_D} E_{\text{train}}(\pi) \tag{1}$$

where $\Pi_D$ is the set of policies used to generate the dataset $D$, $|\Pi_D|$ denotes the number of policies in this set, and $E_{\text{train}}(\pi)$ represents the energy consumed to train the policy $\pi$. As we strive for more equitable comparisons between approaches to training autonomous agents, we urge the research community to consider the cost of acquiring training data. To this end, we release datasets for each domain and task in A2Perf, along with their associated training sample costs. While the specific expertise levels may vary across domains and tasks, we generally consider three categories: `novice`, `intermediate`, and `expert`. See Appendix D for the dataset collection procedure and Appendix E for details on the dataset format.

## 3.2 SYSTEM PERFORMANCE

System metrics provide insight into the feasibility of deploying autonomous agents, particularly considering the scaling demands on energy and data efficiency [15]. A2Perf uses the CodeCarbon library [30] to track metrics during training, such as energy usage, power draw, RAM consumption, and wall-clock time. Energy and power usage inform the user about the sustainability and power costs associated with training the agent, which is particularly important in power-constrained environments or when planning for long-term, continuous training [46]. RAM consumption metrics help in understanding the memory efficiency of the training process, as high RAM consumption may limit the settings where the agent can be trained or require costly hardware upgrades [39]. During the inference phase, A2Perf records power draw, RAM consumption, and average inference time.

System performance measurements may vary significantly across different experimental setups. To ensure reproducibility and facilitate meaningful comparisons, we strongly recommend that users report the deep learning framework, CPU model, GPU model, and Python version used when running A2Perf. Providing this information allows for more accurate interpretation of results. For details on our experimental setup, please refer to Appendix B.

## 3.3 RELIABILITY

| Phase | Metric Name | Description | Equation |
|---|---|---|---|
| Training | Dispersion Within Runs | Measures higher-frequency variability using IQR within a sliding window along the detrended training curve. Lower values indicate more stable performance. | $\frac{1}{T-4} \sum_{t=3}^{T-2} \mathrm{IQR}\left(\{\Delta P_{t'}\}_{t'=t-2}^{t+2}\right)$ |
| | Short-term Risk (CVaR) | Estimates extreme short-term performance drops. Lower values indicate less risk of sudden drops. | $\mathrm{CVaR}_\alpha \left(\Delta P_t\right)_{t=1}^T$ |
| | Long-term Risk (CVaR) | Captures potential for long-term performance decrease. Lower values indicate less risk of degradation. | $\mathrm{CVaR}_\alpha \left(\max_{t' \leq t} P_{t'} - P_t\right)$ |
| | Dispersion Across Runs | Measures variance across training runs. Lower values indicate more consistent performance across runs. | $\frac{1}{T} \sum_{t=1}^T \mathrm{IQR}\left(\{P_{t,j}\}_{j=1}^n\right)$ |
| | Risk Across Runs (CVaR) | Measures expected performance of worst-performing agents. Higher values indicate better worst-case performance. | $\mathrm{CVaR}_\alpha \left(P_{T,j}\right)_{j=1}^n$ |
| Inference | Dispersion Across Rollouts | Measures variability in performance across multiple rollouts. Lower values indicate more consistent performance. | $\mathrm{IQR}\left(R_i\right)_{i=1}^m$ |
| | Risk Across Rollouts (CVaR) | Measures worst-case performance during inference. Higher values indicate better worst-case performance. | $\mathrm{CVaR}_\alpha \left(R_i\right)_{i=1}^m$ |

**Table 3:** Reliability Metrics with Mathematical Formulations. $P_t$: performance at time $t$. $P_{t,j}$: performance at time $t$ for run $j$. $R_i$: performance during rollout $i$. $\Delta P_t = P_t - P_{t-1}$: performance change between consecutive time steps (detrended value). $\mathrm{CVaR}_\alpha$: Conditional Value at Risk at level $\alpha$. IQR: Inter-Quartile Range. Sliding window length is 5 time steps centered on $t$, calculated over all $t$ from 3 to $T - 2$ to ensure the window is valid. $T$: total number of time steps. $n$: number of runs (10 for our experiments). $m$: number of rollouts (100 for our experiments).

Reliability signifies safety, accountability, reproducibility, stability, and trustworthiness [8; 51]. A2Perf uses the statistical methods proposed by Chan et al. [8] to measure the reliability of autonomous agents during training and inference. During training, A2Perf examines dispersion across multiple training runs, dispersion over time within a single run, risk across runs, and risk over time. These metrics provide insights into the variability and worst-case performance of the agent. For inference, A2Perf measures dispersion and risk across rollouts to assess the consistency and potential suboptimal performance of the final trained agent. Table 3 provides an overview of the reliability metrics tracked by A2Perf, along with how they should be interpreted. For a detailed description of each metric and their calculation, please refer to the work by Chan et al. [8].

## 3.4 APPLICATION PERFORMANCE

Application performance is measured using task performance and generalization. Task performance is the agent's mean returns when rolled out for 100 episodes on the task it was trained for. Generalization assesses the agent's ability to adapt to tasks outside of its specific training distribution, and is computed as the sum of mean returns for all tasks, including the task the agent was trained to perform.

## 3.5 USING A2PERF METRICS IN PRACTICE

The metrics provided by A2Perf across data cost, application performance, system performance, and reliability offer a holistic view of autonomous agent performance. However, the relative importance of these metrics can vary significantly depending on the specific application domain. For instance, in resource-constrained environments, system performance metrics may be critical, while in safety-critical applications, reliability metrics might take precedence. In Section 5, we demonstrate how these metrics can be applied and interpreted in the context of our three benchmark domains: computer chip floorplanning, web navigation, and quadruped locomotion.

## 4 A2PERF DOMAINS

The domains in A2Perf were selected based on their demonstrated transfer from simulated environments to the real world. The circuit training domain was used in creating an iteration of Google's

| Real-World Challenges | Chip Floorplanning | Web Navigation | Quadruped Locomotion |
|---|:---:|:---:|:---:|
| **(RW1)**\* Training offline from fixed logs. | ✓ | ✓ | ✓ |
| **(RW2)** Learning on the real system from limited samples. | ✗ | ✗ | ✓ |
| **(RW3)** High-dimensional and continuous state and action spaces. | ✓ | ✗ | ✓ |
| **(RW4)** Safety constraints. | ✗ | ✓ | ✓ |
| **(RW5)** Tasks are partially observable, non-stationary or stochastic. | ✗ | ✗ | ✓ |
| **(RW6)** Unspecified, multi-objective or risk sensitive reward functions. | ✓ | ✓ | ✓ |
| **(RW7)** Need for explainable policies. | ✗ | ✓ | ✗ |
| **(RW8)** Real-time inference at the control frequency of the system. | ✗ | ✓ | ✓ |
| **(RW9)** Delays in actuators, sensors or rewards. | ✗ | ✓ | ✓ |

**Table 4:** Real-World Challenges proposed by Dulac-Arnold et al. [14]. Checkmarks (✓) indicate challenges commonly encountered in the general domain area, while (✗) denotes challenges less frequently encountered. The challenge marked with an asterisk (*), RW1, applies to all A2Perf domains, as learning from offline data is possible for all environments. Each broad challenge is encountered in at least one of the A2Perf domain areas, highlighting the relevance of the selected domains to current real-world reinforcement learning problems.

Tensor Processing Unit (TPU) [43]. The quadruped locomotion domain has been shown to transfer successfully to real Unitree Laikago robots [49]. The web navigation domain is derived from Mini-Wob [55], MiniWob++ [40], and gMiniWob [23], and operates in an actual Google Chrome browser, mirroring real-life web interactions. Additionally, [22] showed that policies trained in MiniWob++ transfer to real-life web pages for task completion.

By focusing on domains with demonstrated real-world applicability, progress made within the A2Perf benchmark can directly contribute to improving the performance of downstream real-world (RW) tasks. We specify how each domain aligns with the real-world challenges presented by Dulac-Arnold et al. [14] (Table 4), and denote which of A2Perf's metric categories are important for each domain.

## 4.1 CIRCUIT TRAINING (RW1, RW3, RW6)

Chip floorplanning involves creating a physical layout for a microprocessor, a task that has resisted automation for decades and requires months of human engineering effort. To address this challenge, Google has made Circuit Training available as an open-source framework that uses RL to generate chip floorplans [20]. In this domain, an agent places macros (reusable blocks of circuitry) onto the chip canvas, with the objective of optimizing wirelength, congestion, and density. Even though the state and action spaces are discrete, the number of states and actions increases combinatorially with the number of nodes and cells on the chip (RW3). As an illustration, Mirhoseini et al. [43] calculate that placing 1,000 clusters of nodes on a grid with 1,000 cells results in a state space on the order of $10^{2,500}$, which is vastly larger than the state space of Go at $10^{360}$. Chip design also involves optimizing for multiple objectives, such as maximizing clock frequency, reducing power consumption, and minimizing chip area (RW6). During training, these objectives are approximated using proxy metrics. However, evaluating the true objectives requires time-consuming simulations with industry-grade placement tools [5]. If the results are unsatisfactory, the proxy metrics must be adjusted, and the agents must be retrained, leading to a costly iterative and resource-intensive process.

The metric categories included in A2Perf that are crucial to evaluating Circuit Training agents are **task performance** (optimality of macro placements), **inference reliability** (to ensure consistent macro placements for human designers to build on top of), **inference system performance** (to collaborate with human designers in a timely manner), **generalization** (to optimally place macros unseen netlists),

---

[5]For example, Cadence Innovus and Synopsys IC Compiler

| Ariane (Training) | | BC | DDQN | PPO |
|---|---|---|---|---|
| **Category** | **Metric Name** | | | |
| Data Cost | Training Sample Cost | 48.28 | 0 | 0 |
| Application | Generalization (100 eps. [all tasks]) | -2.18 | -2.19 | -2.05 |
| | Returns (100 eps.) | $-1.10 \pm 0.04$ | $-1.13 \pm 0.04$ | $-0.99 \pm 7.25\text{e-}03$ |
| Reliability | Dispersion Across Runs (IQR) | N/A | $0.03 \pm 0.03$ | $0.04 \pm 0.02$ |
| | Dispersion Within Runs (IQR) | N/A | $0.02 \pm 0.03$ | $4.77\text{e-}03 \pm 4.92\text{e-}03$ |
| | Long Term Risk (CVaR) | N/A | 1.20 | 0.03 |
| | Risk Across Runs (CVaR) | N/A | -1.17 | -1.03 |
| | Short Term Risk (CVaR) | N/A | 0.07 | 0.01 |
| System | Energy Consumed (kWh) | $0.11 \pm 6.45\text{e-}04$ | $108.20 \pm 4.29$ | $120.53 \pm 2.78$ |
| | GPU Power Usage (W) | $211.35 \pm 16.76$ | $585.98 \pm 172.50$ | $692.94 \pm 120.08$ |
| | Mean RAM Usage (GB) | $4.72 \pm 0.53$ | $849.37 \pm 64.85$ | $834.05 \pm 55.90$ |
| | Peak RAM Usage (GB) | $5.25 \pm 0.07$ | $889.56 \pm 23.44$ | $906.45 \pm 68.01$ |
| | Wall Clock Time (Hours) | $0.48 \pm 2.61\text{e-}03$ | $21.94 \pm 0.90$ | $23.95 \pm 0.54$ |
| Ariane (Inference) | | | | |
| Reliability | Dispersion Across Rollouts (IQR) | 0.01 | 0.05 | 0.01 |
| | Risk Across Rollouts (CVaR) | -1.23 | -1.25 | -1.01 |
| System | GPU Power Usage (W) | $136.91 \pm 21.48$ | $69.50 \pm 4.60$ | $49.43 \pm 30.29$ |
| | Inference Time (ms) | $10.0 \pm 0.46$ | $20.0 \pm 2.69$ | $20.0 \pm 2.68$ |
| | Mean RAM Usage (GB) | $2.19 \pm 0.21$ | $2.15 \pm 0.30$ | $2.51 \pm 0.49$ |
| | Peak RAM Usage (GB) | $2.29 \pm 0.01$ | $2.28 \pm 0.13$ | $2.71 \pm 0.62$ |

**Table 5:** Metrics for the Ariane Netlist task of CircuitTraining-v0. All metrics are averaged over ten random seeds. We report mean and standard deviation for metrics where it is applicable. BC results are obtained by training on the entire `intermediate` dataset.

and **data cost** (due to many netlists being proprietary and the high overhead of human designers producing final macro placements).

## 4.2 WEB NAVIGATION (RW1, RW4, RW6, RW7, RW8, RW9)

Software tools exist to automate browser tasks[6], but due to the varied formatting of websites, hand-crafted algorithms are not a viable solution for general web navigation. Researchers have begun applying learning algorithms to design agents that can understand web pages [24] and automatically navigate through them to fill out forms [23; 22]. In A2Perf, we use gMiniWob [23] to create mock websites that act as environments for the agent. See Appendix F for details about the website genera-tion process and agent interaction. To achieve maximum rewards, the agent must avoid malicious links and advertisement banners (RW4) while correctly filling out all fields in web forms. The combination of these constraints create a multi-objective reward function (RW6). The explainability of an agent's decision-making is also important, particularly when agents handle sensitive tasks such as online shopping or investing (RW7). Finally, agents must be robust to the system challenges of real-time inference, such as inference speed and network delays (RW8, RW9).

The metric categories included in A2Perf that are crucial to evaluating web navigation agents are **task performance** (general correctness of form-filling), **inference system performance** (for seamlessly navigating the web at speeds similar to humans), **inference reliability** (to avoid dangerous actions like clicking malicious links), **generalization** (to handle varying website designs), and **training system performance** (to account for the computational demands of training on diverse web environments, which often requires tokenizing HTML web pages).

## 4.3 QUADRUPED LOCOMOTION (RW1, RW2, RW3, RW4, RW5, RW6, RW8, RW9)

In recent years, the robotics community has gradually shifted towards training autonomous agents for robotic control. A prominent example of this trend is seen in quadruped locomotion, where RL has become the dominant technique. We followed the pioneering work of Peng et al. [49], in which a quadruped robot learns complex locomotion skills such as pacing, trotting, spinning, hop-turning, and side-stepping by imitating motion capture data from a real dog.

---

[6]Selenium, used in A2Perf, is a popular browser automation tool.

| Difficulty 1, 1 Website (Training) | | | | |
|---|---|---|---|---|
| | | **BC** | **DDQN** | **PPO** |
| **Category** | **Metric Name** | | | |
| System | Energy Consumed (kWh) | $0.04 \pm 6.02 \times 10^{-4}$ | $29.56 \pm 7.23$ | $28.82 \pm 1.19$ |
| | GPU Power Usage (W) | $125.89 \pm 2.53$ | $265.09 \pm 21.50$ | $305.15 \pm 34.41$ |
| | Mean RAM Usage (GB) | $4.10 \pm 0.33$ | $1140.98 \pm 580.55$ | $1592.45 \pm 388.64$ |
| | Peak RAM Usage (GB) | $4.23 \pm 0.04$ | $1931.54 \pm 242.31$ | $2305.57 \pm 135.48$ |
| | Wall Clock Time (Hours) | $0.31 \pm 4.91 \times 10^{-3}$ | $8.13 \pm 5.17$ | $10.50 \pm 0.44$ |
| **Difficulty 1, 1 Website (Inference)** | | | | |
| System | GPU Power Usage (W) | $108.61 \pm 15.76$ | $59.61 \pm 1.41$ | $60.26 \pm 1.14$ |
| | Inference Time (ms) | $3.07 \pm 0.47$ | $110 \pm 9.93$ | $120 \pm 9.71$ |
| | Mean RAM Usage (GB) | $1.97 \pm 0.32$ | $2.08 \pm 0.20$ | $2.12 \pm 0.17$ |
| | Peak RAM Usage (GB) | $2.11 \pm 0.11$ | $2.18 \pm 0.11$ | $2.19 \pm 0.09$ |

**Table 6:** Metrics for the "Difficulty Level 1, 1 Website" task of WebNavigation-v0. All metrics are averaged over ten random seeds. We report mean and standard deviation for metrics where it is applicable. BC results are obtained by training on the entire `novice` dataset.

Given the physical dynamics involved in quadruped locomotion, research often necessitates learning directly from limited samples on the actual robot (RW2). Learning walking gaits also involves high-dimensional, continuous state and action spaces (RW3), as the robot needs to precisely control multiple joints and limbs to navigate complex environments. The agent must reason about complex dynamics, avoid unsafe falls (RW4), adapt gaits to various speeds and terrains (RW5), and operate in partially observable environments (RW5) where states like contact forces are not directly measurable. Optimizing robotic controllers is usually multi-objective (RW6), balancing competing objectives like locomotion speed, stability, satisfying safety constraints, and minimizing energy expenditure. Furthermore, real-time inference (RW8) and dealing with system delays (RW9) are critical for controlling robots, as slow computations or delays can negatively impact stability and performance.

The metric categories included in A2Perf that are crucial to evaluating quadruped locomotion agents are **task performance** (accuracy in imitating desired gaits), **inference reliability** (to ensure smooth, stable walking without sudden dangerous movements), **inference system performance** (for real-time responsiveness and energy efficiency on onboard compute), **generalization** (to adapt to novel terrains and morphologies of the robot).

## 5 EVALUATION

We show how A2Perf can aid algorithm development and evaluation on challenging, real-world problems. We highlight A2Perf's evaluation capabilities along the axes of training sample cost, system performance, and reliability. For all domains and tasks, results are averaged over ten random seeds to ensure robustness and reproducibility. See Appendix A for more experimental results.

### 5.1 COMPARING ACROSS ALGORITHM TYPES WITH DATA COST

A2Perf provides datasets generated with agents of varying expertise (Section 3.1), along with their associated training sample costs. This enables the comparison of agents by considering both task performance and the cost of acquiring training data, which can vary significantly across different approaches like IL and RL. Our experiments in the chip floorplanning domain show that while behavioral cloning's (BC) performance is competitive with DDQN and PPO (Table 7), the training sample cost (average energy consumed to train an agent that generates the data) was 48.28 kWh.

Furthermore, this formulation allows researchers to combine the training sample cost with the energy consumed during training for offline, online, or hybrid methods, providing a total energy cost that can be directly compared. For example, the offline training of the BC agent for the Ariane netlist consumed 0.11 kWh. Therefore, the total energy cost for the BC agent would be 48.39 kWh (48.28 kWh for generating the offline data + 0.11 kWh for offline training). This total energy cost can then be compared with the energy consumed by online methods like DDQN and PPO, which amounted to 108.20 kWh and 120.53 kWh, respectively (Table 10). In the case of a hybrid method that uses both offline data and online training, the total energy cost would be calculated by adding the training sample cost for the offline data to the energy consumed during the online training phase.

| Ariane (Training) | | | | |
|---|---|---|---|---|
| | | **BC** | **DDQN** | **PPO** |
| **Category** | **Metric Name** | | | |
| Data Cost | Training Sample Cost | 48.28 | 0 | 0 |
| System | Energy Consumed (kWh) | $0.11 \pm 6.45e\text{-}04$ | $108.20 \pm 4.29$ | $120.53 \pm 2.78$ |
| | GPU Power Usage (W) | $211.35 \pm 16.76$ | $585.98 \pm 172.50$ | $692.94 \pm 120.08$ |
| Ariane (Inference) | | | | |
| Reliability | Dispersion Across Rollouts (IQR) | 0.01 | 0.05 | 0.01 |
| | Risk Across Rollouts (CVaR) | -1.23 | -1.25 | -1.01 |
| System | GPU Power Usage (W) | $136.91 \pm 21.48$ | $69.50 \pm 4.60$ | $49.43 \pm 30.29$ |
| | Inference Time (ms) | $10.0 \pm 0.46$ | $20.0 \pm 2.69$ | $20.0 \pm 2.68$ |
| | Mean RAM Usage (GB) | $2.19 \pm 0.21$ | $2.15 \pm 0.30$ | $2.51 \pm 0.49$ |
| | Peak RAM Usage (GB) | $2.29 \pm 0.01$ | $2.28 \pm 0.13$ | $2.71 \pm 0.62$ |

**Table 7:** Metrics for the Ariane Netlist task of CircuitTraining-v0. All metrics are averaged over ten random seeds. We report mean and standard deviation for metrics where it is applicable. BC results are obtained by training on the entire `intermediate` dataset.

## 5.2 SYSTEM PERFORMANCE FOR TRAINING AND DEPLOYMENT FEASIBILITY

Our experiments in the web navigation domain highlight the importance of considering hardware constraints and performance requirements of autonomous agents. During training, PPO agents had a peak RAM usage of $2.3 \pm 0.14$ TB (Table 6). This high memory footprint can be attributed to the need for distributed experiments running hundreds of Google Chrome processes and storing batches of data, which involves tokenizing the entire DOM[7] tree of HTML elements on each web page. Such memory demands can limit the accessibility of training agents, as not all researchers may have access to the necessary hardware resources. To put this into perspective, training a variant of the GPT-3 language model with approximately 72 billion parameters would require a similar amount of memory, assuming each parameter is stored as a 32-bit floating-point number [7].

However, the resource usage of these agents becomes more manageable for deployment. The 120 ms inference time, when combined with the median round-trip latency of ∼68 ms for a 5G network [52], results in a total latency of ∼200 ms. This combined latency is still faster than the average human reaction time of ∼273 ms[8], enabling real-time responsiveness during web navigation tasks. Furthermore, the peak RAM usage of $2.19 \pm 0.09$ GB (Table 6) indicates the feasibility of deploying trained agents directly on consumer-grade devices, such as smartphones, though the inference time may be slower on-device.

## 5.3 ROBUST EVALUATION WITH RELIABILITY METRICS

Computer chip designers using autonomous agents rely on the agent to generate initial placements that they can build upon, so minimizing variability in the agent's performance is crucial. As shown in Table 7, the PPO algorithm exhibited lower dispersion across rollouts (IQR of $0.01$) compared to DDQN (IQR of $0.05$), indicating that PPO is approximately 5x more stable than DDQN when rolling out fixed, trained policies. This suggests that PPO would provide more consistent starting points for designers, enabling them to focus on refining and optimizing the floorplan instead of repeatedly rolling out the same policy to get similar initial placements. Additionally, PPO demonstrated lower risk across rollouts (CVaR of $-1.01$) compared to DDQN (CVaR of $-1.25$), indicating that in the worst-performing rollouts, PPO performs about 1.2x better than DDQN on average, reducing the likelihood of designers starting with poor floorplans that require extensive manual adjustments.

In analyzing the "Dog Pace" task of QuadrupedLocomotion-v0 (Table 8), we observe overlapping error bars on the returns for PPO and SAC. To better understand their tradeoffs, we use the reliability metrics. PPO provides a 2x reduction in both short-term and long-term risks compared to SAC, making PPO more stable. This stability potentially makes PPO a safer option for training quadrupeds

---

[7]https://en.wikipedia.org/wiki/Document_Object_Model
[8]https://humanbenchmark.com/tests/reactiontime/statistics

| Dog Pace (Training) | | BC | PPO | SAC |
|---|---|---|---|---|
| **Category** | **Metric Name** | | | |
| Application | Generalization (100 eps. [all tasks]) | 3.99 | 3.36 | 5.03 |
| | Returns (100 eps.) | $7.00 \pm 4.68$ | $9.94 \pm 15.59$ | $6.96 \pm 6.72$ |
| Reliability | Dispersion Across Runs (IQR) | N/A | $9.63 \pm 7.27$ | $3.61 \pm 3.88$ |
| | Dispersion Within Runs (IQR) | N/A | $2.22 \pm 1.97$ | $2.98 \pm 3.64$ |
| | Long Term Risk (CVaR) | N/A | 13.00 | 25.82 |
| | Risk Across Runs (CVaR) | N/A | 13.74 | 8.55 |
| | Short Term Risk (CVaR) | N/A | 5.81 | 10.19 |
| **Dog Pace (Inference)** | | | | |
| Reliability | Dispersion Across Rollouts (IQR) | 0.52 | 8.76 | 4.80 |
| | Risk Across Rollouts (CVaR) | 0.33 | 0.46 | 1.69 |

**Table 8:** Metrics for the "dog pace" gait of QuadrupedLocomotion-v0, averaged over ten random seeds. We report mean and standard deviation for metrics where it is applicable. BC results are obtained by training on the entire `expert` dataset.

in the real world, where less sporadic behavior is needed. Conversely, SAC performs 3.7x better than PPO in the worst-case rollouts on average and demonstrates a 1.8x improvement in dispersion across rollouts, indicating more consistent gaits during deployment – essential from a safety perspective.

## 6 LIMITATIONS AND FUTURE WORK

A2Perf includes three domains that cover a diverse range of real-world applications and challenges, but there is room for expansion to a wider range of tasks. Thanks to A2Perf's integration with Gymnasium [59] (previously OpenAI Gym) and the implementation of baselines using TF-Agents [19], adding new domains and baselines is straightforward, making it easy for researchers to contribute to the platform.

Future work could expand A2Perf to include multi-agent domains and tasks, reflecting real-world scenarios where autonomous agents interact with other agents and humans. Additionally, adding support for measuring system performance on customized hardware platforms would provide more precise insights into performance in target deployment environments, as current evaluations are primarily conducted on desktop and server machines. Another area of future work is further standardizing evaluations in A2Perf, addressing potential variations due to different hardware setups, Python versions, and code implementations. These efforts will enhance reproducibility and facilitate more accurate comparisons across different research environments, further solidifying A2Perf's role as a comprehensive benchmark for real-world autonomous agents.

## 7 CONCLUSION

We need more holistic metrics and representative benchmarks to measure progress. To this end, we introduced A2Perf, a benchmarking suite that can be used for evaluating autonomous agents on challenging tasks from domains such as computer chip floorplanning, web navigation, and quadruped locomotion. A2Perf provides a standardized set of metrics across data cost, application performance, system resource efficiency, and reliability, enabling a comprehensive comparison of different algorithms. Our evaluations demonstrate A2Perf's effectiveness in identifying the strengths and weaknesses of various approaches to developing autonomous agents. We encourage the community to contribute new domains, tasks, and algorithms to A2Perf, making it an even more comprehensive platform for benchmarking autonomous agents in real-world-inspired settings.

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

# Part I

# Appendix

## A  ADDITIONAL EXPERIMENTS

We present an extensive set of additional experiments that showcase A2Perf's capabilities in evaluating autonomous agents across various domains and tasks. The results encompass a wide range of metrics, including data cost, reliability, system performance, and application performance, providing a holistic view of the strengths and limitations of different algorithmic approaches.

The circuit training domain experiments (Appendix A.1) reveal interesting trade-offs between behavioral cloning, DDQN, and PPO in terms of data efficiency, computational requirements, and performance consistency. Moving to the quadruped locomotion domain (Appendix A.2), we observe how the reliability metrics shed light on the robustness and worst-case behavior of the agents during both training and inference phases. The web navigation domain (Appendix A.2) introduces an additional layer of complexity, with websites of varying difficulty levels. Here, the system performance metrics highlight the substantial computational demands, particularly in terms of memory usage, associated with training web navigation agents. To further facilitate a clear and intuitive comparison of the algorithms' performance across all domains and tasks, we have included graphical visualizations (Appendix A.4) that summarize the key metrics along different evaluation dimensions.

These experiments show A2Perf's versatility in providing a comprehensive and nuanced evaluation of autonomous agents operating in diverse and realistic settings. By considering multiple performance aspects and presenting the results in both tabular and graphical formats, A2Perf enables researchers and practitioners to gain valuable insights into the behavior and limitations of different algorithmic choices, ultimately guiding the development of more robust and efficient autonomous agents.

### A.1  CIRCUIT TRAINING

This section shows the full set of metrics for the toy macro standard cell and Ariane netlists in the circuit training domain. The results highlight the differences in data cost, reliability, system performance, and application performance between behavioral cloning (BC), DDQN, and PPO.

| Toy Macro Standard Cell (Training) | | | | |
|---|---|---|---|---|
| | | **BC** | **DDQN** | **PPO** |
| **Category** | **Metric Name** | | | |
| Data Cost | Training Sample Cost (kWh) | 4.44 | 0 | 0 |
| Application | Generalization (100 eps. [all tasks]) | -2.19 | -2.20 | -2.13 |
| | Returns (100 eps.) | $-0.97 \pm 2.27 \times 10^{-3}$ | $-1.05 \pm 0.04$ | $-0.97 \pm 8.09 \times 10^{-3}$ |
| Reliability | Dispersion Across Runs (IQR) | N/A | $0.01 \pm 0.01$ | $9.07e\text{-}03 \pm 6.43 \times 10^{-3}$ |
| | Dispersion Within Runs (IQR) | N/A | $8.80 \times 10^{-3} \pm 0.01$ | $2.51 \times 10^{-3} \pm 3.61 \times 10^{-3}$ |
| | Long Term Risk (CVaR) | N/A | 1.10 | 0.04 |
| | Risk Across Runs (CVaR) | N/A | -1.08 | -0.99 |
| | Short Term Risk (CVaR) | N/A | 0.03 | $9.89 \times 10^{-3}$ |
| System | Energy Consumed (kWh) | $0.02 \pm 1.97 \times 10^{-4}$ | $5.55 \pm 2.03$ | $15.37 \pm 3.79$ |
| | GPU Power Usage (W) | $188.20 \pm 21.98$ | $448.00 \pm 200.41$ | $307.05 \pm 69.75$ |
| | Peak RAM Usage (GB) | $4.71 \pm 0.02$ | $525.99 \pm 205.64$ | $675.26 \pm 45.30$ |
| | Wall Clock Time (Hours) | $0.10 \pm 1.36 \times 10^{-3}$ | $0.29 \pm 0.57$ | $1.79 \pm 2.16$ |
| Toy Macro Standard Cell (Inference) | | | | |
| Reliability | Dispersion Across Rollouts (IQR) | $1.68 \times 10^{-3}$ | 0.09 | $2.43 \times 10^{-3}$ |
| | Risk Across Rollouts (CVaR) | -0.97 | -1.10 | -0.99 |
| System | GPU Power Usage (W) | $104.97 \pm 22.85$ | $59.45 \pm 1.43$ | $58.97 \pm 1.14$ |
| | Inference Time (ms) | $8.93 \pm 0.51$ | $20 \pm 2.69$ | $20 \pm 2.67$ |
| | Mean RAM Usage (GB) | $1.92 \pm 0.42$ | $1.45 \pm 0.48$ | $1.99 \pm 0.30$ |
| | Peak RAM Usage (GB) | $2.14 \pm 0.03$ | $2.10 \pm 0.05$ | $2.16 \pm 0.07$ |

**Table 9:** Metrics for the "Toy Macro" netlist task of CircuitTraining-v0. All metrics are averaged over ten random seeds.

| Ariane (Training) | | | | |
|---|---|---|---|---|
| | | **BC** | **DDQN** | **PPO** |
| **Category** | **Metric Name** | | | |
| Data Cost | Training Sample Cost | 48.28 | 0 | 0 |
| Application | Generalization (100 eps. [all tasks]) | -2.18 | -2.19 | -2.05 |
| | Returns (100 eps.) | $-1.10 \pm 0.04$ | $-1.13 \pm 0.04$ | $-0.99 \pm 7.25 \times 10^{-3}$ |
| Reliability | Dispersion Across Runs (IQR) | N/A | $0.03 \pm 0.03$ | $0.04 \pm 0.02$ |
| | Dispersion Within Runs (IQR) | N/A | $0.02 \pm 0.03$ | $4.77 \times 10^{-3} \pm 4.92 \times 10^{-3}$ |
| | Long Term Risk (CVaR) | N/A | 1.20 | 0.03 |
| | Risk Across Runs (CVaR) | N/A | -1.17 | -1.03 |
| | Short Term Risk (CVaR) | N/A | 0.07 | 0.01 |
| System | Energy Consumed (kWh) | $0.11 \pm 6.45 \times 10^{-4}$ | $108.20 \pm 4.29$ | $120.53 \pm 2.78$ |
| | GPU Power Usage (W) | $211.35 \pm 16.76$ | $585.98 \pm 172.50$ | $692.94 \pm 120.08$ |
| | Mean RAM Usage (GB) | $4.72 \pm 0.53$ | $849.37 \pm 64.85$ | $834.05 \pm 55.90$ |
| | Peak RAM Usage (GB) | $5.25 \pm 0.07$ | $889.56 \pm 23.44$ | $906.45 \pm 68.01$ |
| | Wall Clock Time (Hours) | $0.48 \pm 2.61e\text{-}03$ | $21.94 \pm 0.90$ | $23.95 \pm 0.54$ |
| Ariane (Inference) | | | | |
| Reliability | Dispersion Across Rollouts (IQR) | 0.01 | 0.05 | 0.01 |
| | Risk Across Rollouts (CVaR) | -1.23 | -1.25 | -1.01 |
| System | GPU Power Usage (W) | $136.91 \pm 21.48$ | $69.50 \pm 4.60$ | $49.43 \pm 30.29$ |
| | Inference Time (ms) | $10.0 \pm 0.46$ | $20.0 \pm 2.69$ | $20.0 \pm 2.68$ |
| | Mean RAM Usage (GB) | $2.19 \pm 0.21$ | $2.15 \pm 0.30$ | $2.51 \pm 0.49$ |
| | Peak RAM Usage (GB) | $2.29 \pm 0.01$ | $2.28 \pm 0.13$ | $2.71 \pm 0.62$ |

**Table 10:** Metrics for the Ariane Netlist task of CircuitTraining-v0. All metrics are averaged over ten random seeds.

A.2  QUADRUPED LOCOMOTION

This section reports the metrics for the dog pace, trot, and spin gaits in the quadruped locomotion domain. The reliability metrics provide insights into the stability and worst-case performance of the algorithms during training and inference.

| Dog Pace (Training) | | | | |
|---|---|---|---|---|
| | | **BC** | **PPO** | **SAC** |
| **Category** | **Metric Name** | | | |
| Data Cost | Training Sample Cost (kWh) | 22.53 | 0 | 0 |
| Application | Generalization (100 eps. [all tasks]) | 3.99 | 3.36 | 5.03 |
| | Returns (100 eps.) | $7.00 \pm 4.68$ | $9.94 \pm 15.59$ | $6.96 \pm 6.72$ |
| Reliability | Dispersion Across Runs (IQR) | N/A | $9.63 \pm 7.27$ | $3.61 \pm 3.88$ |
| | Dispersion Within Runs (IQR) | N/A | $2.22 \pm 1.97$ | $2.98 \pm 3.64$ |
| | Long Term Risk (CVaR) | N/A | 13.00 | 25.82 |
| | Risk Across Runs (CVaR) | N/A | 13.74 | 8.55 |
| | Short Term Risk (CVaR) | N/A | 5.81 | 10.19 |
| System | Energy Consumed (kWh) | $0.11 \pm 0.02$ | $32.46 \pm 0.26$ | $36.22 \pm 2.33$ |
| | GPU Power Usage (W) | $240.64 \pm 5.41$ | $280.12 \pm 23.69$ | $266.37 \pm 9.54$ |
| | Mean RAM Usage (GB) | $3.21 \pm 0.24$ | $532.93 \pm 14.28$ | $516.24 \pm 75.03$ |
| | Peak RAM Usage (GB) | $3.25 \pm 0.01$ | $534.26 \pm 2.04$ | $545.16 \pm 0.50$ |
| | Wall Clock Time (Hours) | $0.46 \pm 0.07$ | $18.73 \pm 0.19$ | $19.41 \pm 2.74$ |
| Dog Pace (Inference) | | | | |
| Reliability | Dispersion Across Rollouts (IQR) | 0.52 | 8.76 | 4.80 |
| | Risk Across Rollouts (CVaR) | 0.33 | 0.46 | 1.69 |
| System | GPU Power Usage (W) | $60.37 \pm 1.78$ | $59.11 \pm 1.31$ | $61.41 \pm 1.96$ |
| | Inference Time (ms) | $2.33 \pm 0.54$ | $2.56 \pm 0.39$ | $2.52 \pm 0.74$ |
| | Mean RAM Usage (GB) | $1.69 \pm 0.31$ | $1.81 \pm 0.14$ | $1.71 \pm 0.30$ |
| | Peak RAM Usage (GB) | $1.82 \pm 0.03$ | $1.84 \pm 9.05\text{e-}03$ | $1.85 \pm 0.04$ |

**Table 11:** Metrics for the "dog pace" gait of QuadrupedLocomotion-v0. All metrics are averaged over ten random seeds

| Dog Trot (Training) | | | | |
|---|---|---|---|---|
| | | **BC** | **PPO** | **SAC** |
| **Category** | **Metric Name** | | | |
| Data Cost | Training Sample Cost (kWh) | 15.77 | 0 | 0 |
| Application | Generalization (100 eps. [all tasks]) | 3.87 | 3.09 | 4.49 |
| | Returns (100 eps.) | $1.06 \pm 0.26$ | $1.49 \pm 1.02$ | $3.51 \pm 2.88$ |
| Reliability | Dispersion Across Runs (IQR) | N/A | $9.07 \pm 4.93$ | $0.85 \pm 1.29$ |
| | Dispersion Within Runs (IQR) | N/A | $0.82 \pm 0.84$ | $0.93 \pm 1.11$ |
| | Long Term Risk (CVaR) | N/A | 6.79 | 8.46 |
| | Risk Across Runs (CVaR) | N/A | 6.00 | 2.58 |
| | Short Term Risk (CVaR) | N/A | 2.41 | 3.20 |
| System | Energy Consumed (kWh) | $0.12 \pm 0.02$ | $16.82 \pm 0.29$ | $19.17 \pm 0.64$ |
| | GPU Power Usage (W) | $242.12 \pm 7.53$ | $277.71 \pm 23.47$ | $269.18 \pm 10.12$ |
| | Mean RAM Usage (GB) | $3.21 \pm 0.25$ | $535.00 \pm 18.77$ | $535.99 \pm 29.49$ |
| | Peak RAM Usage (GB) | $3.26 \pm 0.01$ | $536.47 \pm 1.98$ | $544.80 \pm 4.39$ |
| | Wall Clock Time (Hours) | $0.46 \pm 0.06$ | $18.57 \pm 0.23$ | $18.99 \pm 6.78$ |
| Dog Trot (Inference) | | | | |
| Reliability | Dispersion Across Rollouts (IQR) | 0.32 | 0.89 | 1.25 |
| | Risk Across Rollouts (CVaR) | 0.63 | 0.36 | 1.33 |
| System | GPU Power Usage (W) | $59.32 \pm 1.08$ | $58.91 \pm 1.28$ | $59.39 \pm 1.23$ |
| | Inference Time (ms) | $2.32 \pm 0.49$ | $2.55 \pm 0.57$ | $2.45 \pm 0.35$ |
| | Mean RAM Usage (GB) | $1.66 \pm 0.33$ | $1.76 \pm 0.25$ | $1.80 \pm 0.17$ |
| | Peak RAM Usage (GB) | $1.82 \pm 8.77 \times 10^{-4}$ | $1.85 \pm 0.02$ | $1.85 \pm 0.03$ |

**Table 12:** Metrics for the "dog trot" gait of QuadrupedLocomotion-v0. All metrics are averaged over ten random seeds.

| Dog Spin (Training) | | | | |
|---|---|---|---|---|
| | | **BC** | **PPO** | **SAC** |
| **Category** | **Metric Name** | | | |
| Data Cost | Training Sample Cost (kWh) | 30.17 | 0 | 0 |
| Application | Generalization (100 eps. [all tasks]) | 3.97 | 2.69 | 4.61 |
| | Returns (100 eps.) | $1.54 \pm 0.42$ | $3.82 \pm 6.22$ | $3.84 \pm 1.46$ |
| Reliability | Dispersion Across Runs (IQR) | N/A | $7.92 \pm 4.60$ | $0.74 \pm 0.76$ |
| | Dispersion Within Runs (IQR) | N/A | $1.00 \pm 1.08$ | $0.84 \pm 1.26$ |
| | Long Term Risk (CVaR) | N/A | 8.88 | 14.37 |
| | Risk Across Runs (CVaR) | N/A | 8.29 | 3.82 |
| | Short Term Risk (CVaR) | N/A | 3.09 | 2.99 |
| System | Energy Consumed (kWh) | $0.10 \pm 0.04$ | $17.42 \pm 0.35$ | $18.88 \pm 0.59$ |
| | GPU Power Usage (W) | $216.72 \pm 68.63$ | $278.38 \pm 22.60$ | $264.46 \pm 9.49$ |
| | Mean RAM Usage (GB) | $3.18 \pm 0.26$ | $534.56 \pm 21.28$ | $531.27 \pm 55.64$ |
| | Peak RAM Usage (GB) | $3.23 \pm 0.08$ | $536.10 \pm 3.03$ | $477.22 \pm 172.63$ |
| | Wall Clock Time (Hours) | $0.45 \pm 0.08$ | $17.13 \pm 6.07$ | $17.02 \pm 9.05$ |
| Dog Spin (Inference) | | | | |
| Reliability | Dispersion Across Rollouts (IQR) | 0.37 | 2.41 | 1.78 |
| | Risk Across Rollouts (CVaR) | 0.28 | 0.12 | 0.55 |
| System | GPU Power Usage (W) | $60.10 \pm 1.14$ | $59.70 \pm 1.22$ | $59.65 \pm 1.73$ |
| | Inference Time (ms) | $2.33 \pm 0.66$ | $2.45 \pm 0.48$ | $2.41 \pm 0.22$ |
| | Mean RAM Usage (GB) | $1.68 \pm 0.32$ | $1.79 \pm 0.22$ | $1.75 \pm 0.26$ |
| | Peak RAM Usage (GB) | $1.82 \pm 0.03$ | $1.85 \pm 0.02$ | $1.84 \pm 0.02$ |

**Table 13:** Metrics for the "dog spin" gait of QuadrupedLocomotion-v0. All metrics are averaged over ten random seeds.

A.3   WEB NAVIGATION

This section details the evaluation on websites of varying difficulty levels in the web navigation domain. The system performance metrics underscore the significant computational requirements, especially in terms of RAM usage, for training web navigation agents.

| Difficulty 1, 1 Website (Training) | | | | |
|---|---|---|---|---|
| | | **BC** | **DDQN** | **PPO** |
| **Category** | **Metric Name** | | | |
| Data Cost | Training Sample Cost (kWh) | 14.15 | 0 | 0 |
| Application | Generalization (100 eps. [all tasks]) | -12.94 | -11.15 | -24.54 |
| | Returns (100 eps.) | $-3.57 \pm 2.80$ | $-7.55 \pm 5.74$ | $-13.45 \pm 0.51$ |
| Reliability | Dispersion Across Runs (IQR) | N/A | $0.73 \pm 0.63$ | $4.20 \pm 1.45$ |
| | Dispersion Within Runs (IQR) | N/A | $0.37 \pm 0.68$ | $0.57 \pm 0.53$ |
| | Long Term Risk (CVaR) | N/A | 9.32 | 12.12 |
| | Risk Across Runs (CVaR) | N/A | -2.75 | -13.11 |
| | Short Term Risk (CVaR) | N/A | 1.79 | 1.86 |
| System | Energy Consumed (kWh) | $0.04 \pm 6.02 \times 10^{-4}$ | $29.56 \pm 7.23$ | $28.82 \pm 1.19$ |
| | GPU Power Usage (W) | $125.89 \pm 2.53$ | $265.09 \pm 21.50$ | $305.15 \pm 34.41$ |
| | Mean RAM Usage (GB) | $4.10 \pm 0.33$ | $1140.98 \pm 580.55$ | $1592.45 \pm 388.64$ |
| | Peak RAM Usage (GB) | $4.23 \pm 0.04$ | $1931.54 \pm 242.31$ | $2305.57 \pm 135.48$ |
| | Wall Clock Time (Hours) | $0.31 \pm 4.91 \times 10^{-3}$ | $8.13 \pm 5.17$ | $10.50 \pm 0.44$ |
| Difficulty 1, 1 Website (Inference) | | | | |
| Reliability | Dispersion Across Rollouts (IQR) | 3.36 | 11.75 | 0.50 |
| | Risk Across Rollouts (CVaR) | -10.65 | -13.25 | -13.75 |
| System | GPU Power Usage (W) | $108.61 \pm 15.76$ | $59.61 \pm 1.41$ | $60.26 \pm 1.14$ |
| | Inference Time (ms) | $3.07 \pm 0.47$ | $110 \pm 9.93$ | $120 \pm 9.71$ |
| | Mean RAM Usage (GB) | $1.97 \pm 0.32$ | $2.08 \pm 0.20$ | $2.12 \pm 0.17$ |
| | Peak RAM Usage (GB) | $2.11 \pm 0.11$ | $2.18 \pm 0.11$ | $2.19 \pm 0.09$ |

**Table 14:** Metrics for "difficulty 1, 1 website" task of WebNavigation-v0. All metrics are averaged over ten random seeds.

| Difficulty 1, 5 Websites (Training) | | | | |
|---|---|---|---|---|
| | | **BC** | **DDQN** | **PPO** |
| **Category** | **Metric Name** | | | |
| Data Cost | Training Sample Cost (kWh) | 13.66 | 0 | 0 |
| Application | Generalization (100 eps. [all tasks]) | -13.34 | -11.03 | -23.86 |
| | Returns (100 eps.) | $-4.87 \pm 3.33$ | $-3.43 \pm 4.58$ | $-12.37 \pm 3.53$ |
| Reliability | Dispersion Across Runs (IQR) | N/A | $0.43 \pm 0.55$ | $3.42 \pm 1.08$ |
| | Dispersion Within Runs (IQR) | N/A | $0.49 \pm 0.97$ | $0.75 \pm 0.55$ |
| | Long Term Risk (CVaR) | N/A | 11.27 | 11.70 |
| | Risk Across Runs (CVaR) | N/A | -1.26 | -12.60 |
| | Short Term Risk (CVaR) | N/A | 2.47 | 2.05 |
| System | Energy Consumed (kWh) | $0.04 \pm 4.82 \times 10^{-4}$ | $31.59 \pm 5.19$ | $28.48 \pm 1.22$ |
| | GPU Power Usage (W) | $126.04 \pm 4.03$ | $265.81 \pm 22.08$ | $303.28 \pm 34.99$ |
| | Mean RAM Usage (GB) | $4.03 \pm 0.34$ | $1206.86 \pm 466.37$ | $1545.56 \pm 427.22$ |
| | Peak RAM Usage (GB) | $4.15 \pm 0.11$ | $1928.69 \pm 209.62$ | $2227.07 \pm 210.77$ |
| | Wall Clock Time (Hours) | $0.30 \pm 3.71 \times 10^{-3}$ | $9.35 \pm 4.70$ | $10.45 \pm 0.31$ |
| Difficulty 1, 5 Websites (Inference) | | | | |
| Reliability | Dispersion Across Rollouts (IQR) | 5.96 | 0.29 | 0.50 |
| | Risk Across Rollouts (CVaR) | -11.36 | -13.46 | -13.75 |
| System | GPU Power Usage (W) | $108.13 \pm 16.85$ | $60.87 \pm 5.78$ | $60.17 \pm 1.67$ |
| | Inference Time (ms) | $3.04 \pm 0.44$ | $110 \pm 9.83$ | $120 \pm 9.21$ |
| | Mean RAM Usage (GB) | $1.97 \pm 0.33$ | $2.07 \pm 0.32$ | $2.12 \pm 0.16$ |
| | Peak RAM Usage (GB) | $2.12 \pm 0.03$ | $2.57 \pm 0.86$ | $2.19 \pm 0.01$ |

**Table 15:** Metrics for "difficulty 1, 5 websites" task of WebNavigation-v0. All metrics are averaged over ten random seeds.

| Difficulty 1, 10 Websites (Training) | | | | |
|---|---|---|---|---|
| | | **BC** | **DDQN** | **PPO** |
| **Category** | **Metric Name** | | | |
| Data Cost | Training Sample Cost (kWh) | 19.71 | 0 | 0 |
| Application | Returns (100 eps.) | -4.68 ± 3.28 | -3.14 ± 4.24 | -12.73 ± 2.86 |
| Reliability | Dispersion Across Runs (IQR) | N/A | 0.32 ± 0.47 | 3.67 ± 0.63 |
| | Dispersion Within Runs (IQR) | N/A | 0.42 ± 0.86 | 0.79 ± 0.53 |
| | Long Term Risk (CVaR) | N/A | 9.47 | 11.85 |
| | Risk Across Runs (CVaR) | N/A | -1.44 | -12.79 |
| | Short Term Risk (CVaR) | N/A | 2.27 | 1.82 |
| System | Energy Consumed (kWh) | $0.05 \pm 2.41 \times 10^{-4}$ | 27.19 ± 11.22 | 20.35 ± 5.77 |
| | GPU Power Usage (W) | 125.88 ± 2.33 | 264.98 ± 24.45 | 304.66 ± 33.77 |
| | Mean RAM Usage (GB) | 3.56 ± 0.39 | 1214.88 ± 524.77 | 1034.85 ± 424.83 |
| | Peak RAM Usage (GB) | 4.10 ± 0.05 | 1784.37 ± 641.82 | 1665.15 ± 395.14 |
| | Wall Clock Time (Hours) | 0.32 ± 1.43e-03 | 7.80 ± 5.20 | 3.10 ± 5.06 |
| Difficulty 1, 10 Websites (Inference) | | | | |
| Reliability | Dispersion Across Rollouts (IQR) | 5.86 | 0.25 | 0.50 |
| | Risk Across Rollouts (CVaR) | -11.33 | -13.28 | -13.75 |
| System | GPU Power Usage (W) | 108.26 ± 16.34 | 59.95 ± 1.49 | 59.67 ± 1.57 |
| | Inference Time (ms) | 3.05 ± 0.45 | 110 ± 8.42 | 120 ± 9.90 |
| | Mean RAM Usage (GB) | 1.97 ± 0.35 | 2.06 ± 0.27 | 2.13 ± 0.16 |
| | Peak RAM Usage (GB) | 2.13 ± 0.04 | 2.17 ± 0.03 | 2.20 ± 0.03 |

**Table 16:** Metrics for "difficulty 1, 10 websites" task of WebNavigation-v0. All metrics are averaged over ten random seeds.

## A.4 RADAR PLOTS FOR EASY VISUAL COMPARISON

These figures provide a graphical representation of the key metrics across all domains and tasks, enabling a visual comparison of the algorithms' performance along the different evaluation axes.

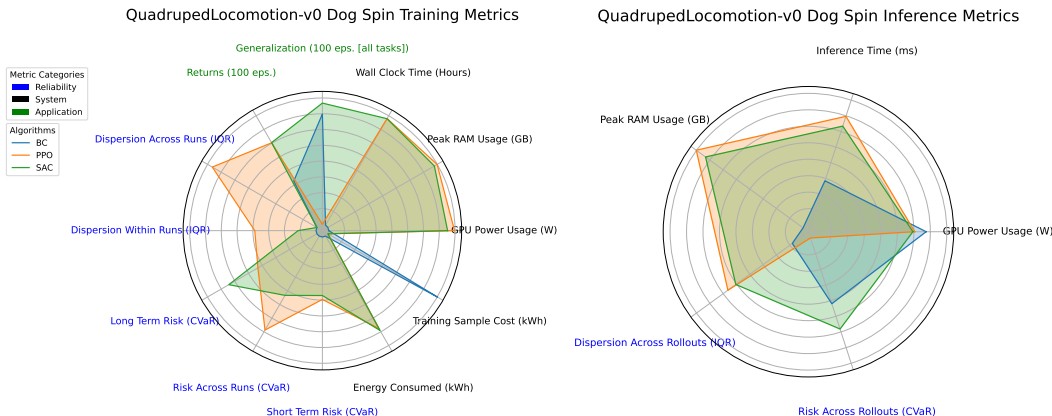

**Figure 1:** Graphical representation of metrics for the "dog spin" gait of QuadrupedLocomotion-v0

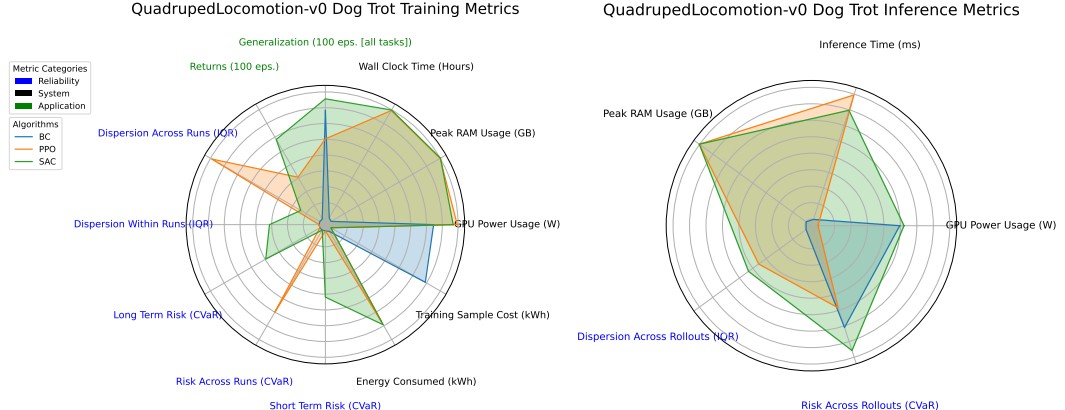

**Figure 2:** Graphical representation of metrics for the "dog trot" gait of QuadrupedLocomotion-v0

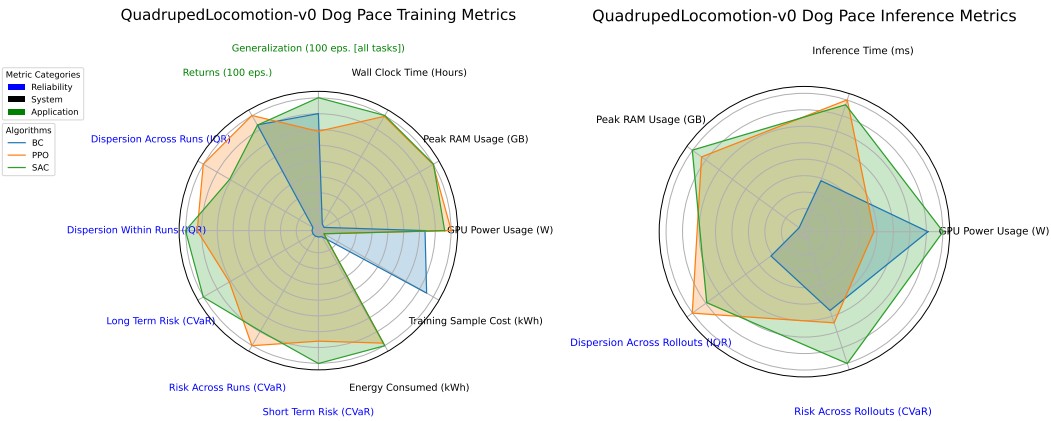

**Figure 3:** Graphical representation of metrics for the "dog pace" gait of QuadrupedLocomotion-v0

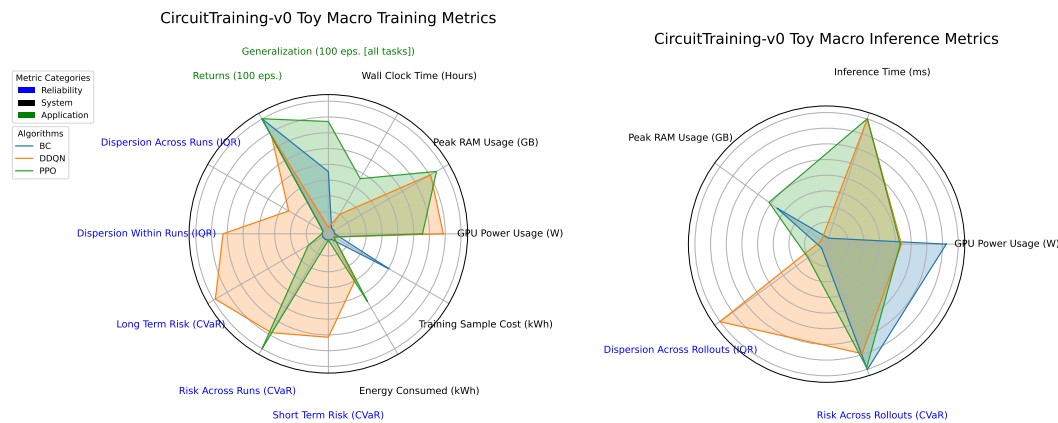

**Figure 4:** Graphical representation of metrics for the "Toy Macro" netlist task of CircuitTraining-v0

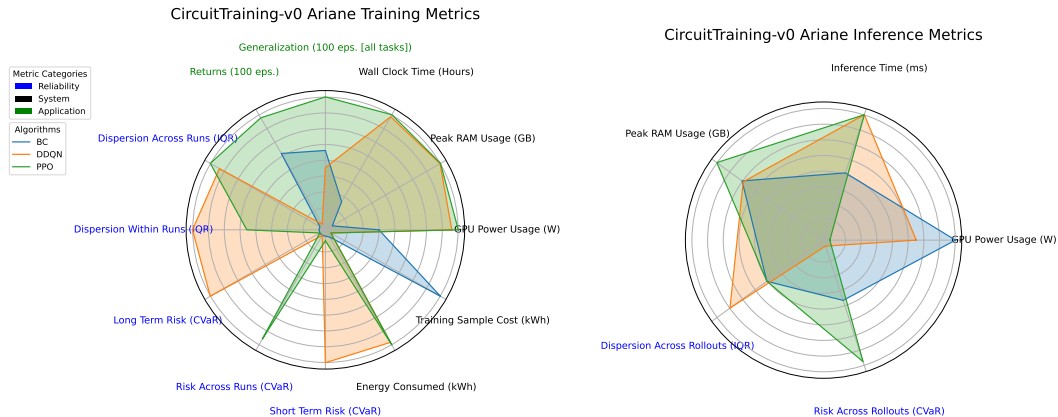

**Figure 5:** Graphical representation of metrics for the "Ariane" netlist task of CircuitTraining-v0

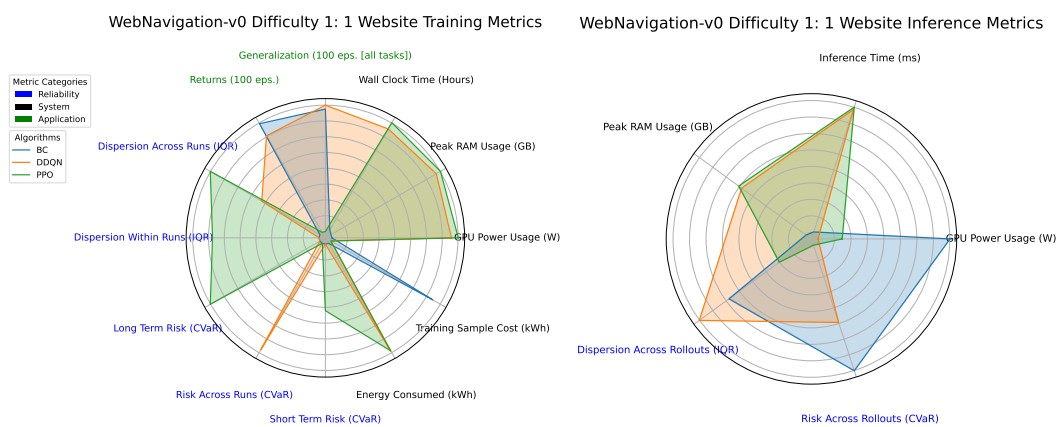

**Figure 6:** Graphical representation of metrics for the "difficulty 1, 1 website" task of WebNavigation-v0

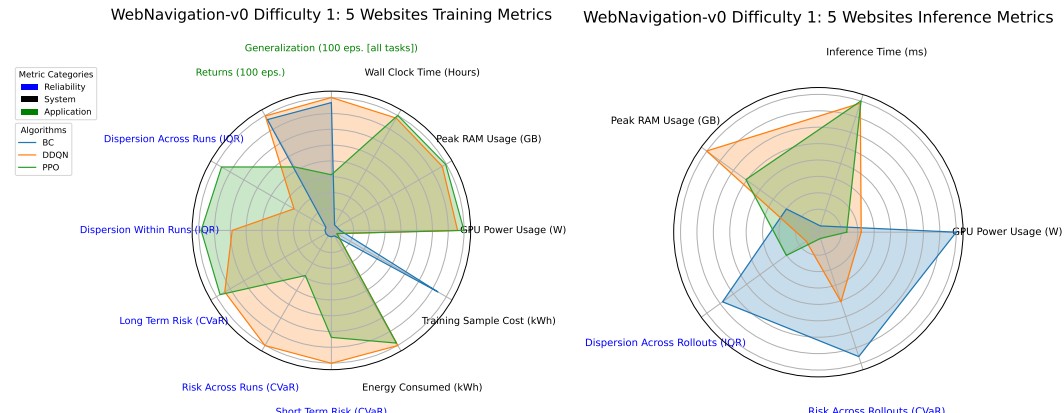

**Figure 7:** Graphical representation of metrics for the "difficulty 1, 5 websites" task of WebNavigation-v0

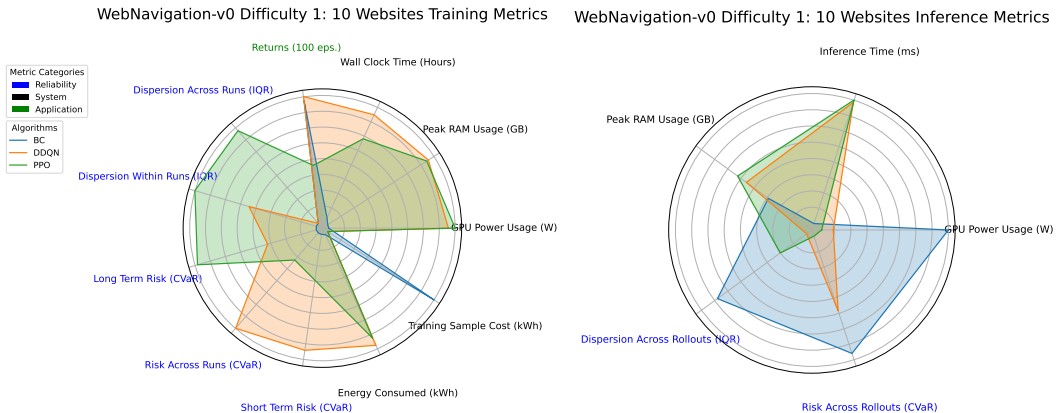

**Figure 8:** Graphical representation of metrics for the "difficulty 1, 10 websites" task of WebNavigation-v0

## B  EXPERIMENTAL SETUP

### B.1  TRAINING

We used the Tensorflow Agents [19] library to conduct distributed reinforcement learning experiments across the three domains: computer chip floorplanning, web navigation, and quadruped locomotion. Our training setup consisted of one training server (a Google Cloud a2-highgpu-8g instance[9]) equipped with four NVIDIA A100 GPUs, and multiple collect servers (Google Cloud n2-standard-96 instances[10]) with 96 vCPUs running in parallel.

The number of collect jobs running simultaneously varied depending on the specific domain and the available resources (such as CPU and memory) on the collect machines, which are important for running the environments efficiently. When using a collect machine with 96 vCPUs, we adjusted the number of environment instances based on the computational requirements of each domain:

1. **Quadruped Locomotion**: With 96 vCPUs on the collect machine, we ran 44 quadruped locomotion environment instances concurrently using Python 3.9.

2. **Computer Chip Floorplanning**: For the computer chip floorplanning domain, we ran 25 computer chip floorplanning environment instances on a collect machine with 96 vCPUs using Python 3.10.

3. **Web Navigation**: When running web navigation experiments on a collect machine with 96 vCPUs, we instantiated 40 web navigation environment instances simultaneously using Python 3.10.

The behavioral cloning experiments for all three domains used the same setup as the online training experiments, with one training server equipped with four A100 GPUs.

### B.2  INFERENCE

For the inference phase, we used a single machine equipped with one NVIDIA V100 GPU to evaluate the trained models across all three domains: computer chip floorplanning, web navigation, and quadruped locomotion. The difference in hardware between the training and inference setups does not affect the application performance metrics, as these metrics are independent of the hardware and reflect the effectiveness of the trained models. However, the system performance metrics, such as inference time and memory usage, may vary depending on the specific hardware used during inference.

---

[9]https://cloud.google.com/compute/docs/gpus#a100-gpus
[10]https://cloud.google.com/compute/docs/general-purpose-machines#n2-standard

## C    HYPERPARAMETERS

| Hyperparameter | BC | PPO | DDQN |
|---|---|---|---|
| **Toy Macro Standard Cell** | | | |
| Batch Size | 64 | 128 | 256 |
| Learning Rate | 1e-4 | 4e-4 | 4e-5 |
| Environment Batch Size | - | 512 | 512 |
| Number of Epochs | - | 6 | - |
| Number of Iterations | 200 | 5000 | 10000 |
| Entropy Regularization | - | 1e-2 | - |
| Number of Episodes Per Iteration | - | 32 | - |
| Epsilon Greedy | - | - | 0.3 |
| Replay Buffer Capacity | - | - | 1000000 |
| **Ariane** | | | |
| Batch Size | 64 | 128 | 256 |
| Learning Rate | 1e-4 | 4e-4 | 4e-5 |
| Environment Batch Size | - | 512 | 512 |
| Number of Epochs | - | 4 | - |
| Number of Iterations | 200 | 250 | 100000 |
| Entropy Regularization | - | 1e-2 | - |
| Number of Episodes Per Iteration | - | 1024 | - |
| Epsilon Greedy | - | - | 0.3 |
| Replay Buffer Capacity | - | - | 10000000 |

**Table 17:** Circuit Training Hyperparameters

| Hyperparameter | BC | PPO | DDQN |
|---|---|---|---|
| Batch Size | 128 | 128 | 128 |
| Learning Rate | 1e-4 | 3e-6 | 3e-6 |
| Entropy Regularization | - | 1e-2 | - |
| Number of Episodes Per Iteration | - | 512 | - |
| Environment Batch Size | - | 512 | 512 |
| Number of Epochs | - | 4 | - |
| Number of Iterations | 5000 | 200 | 50000 |
| Epsilon Greedy | - | - | 0.3 |
| Replay Buffer Capacity | - | - | 1000000 |
| Maximum Vocabulary Size | 500 | 500 | 500 |
| Latent Dimension | 50 | 50 | 50 |
| Embedding Dimension | 100 | 100 | 100 |
| Profile Value Dropout | 1.0 | 1.0 | 1.0 |

**Table 18:** Web Navigation Hyperparameters

| Hyperparameter | BC | PPO | SAC |
|----------------|-----|------|---------|
| Batch Size | 64 | 128 | 256 |
| Learning Rate | 1e-4 | 1e-5 | 3e-4 |
| Environment Batch Size | - | 512 | 512 |
| Number of Epochs | - | 4 | - |
| Number of Iterations | 1000 | 8000 | 2000000 |
| Entropy Regularization | - | 1e-2 | - |
| Number of Episodes Per Iteration | - | 512 | - |
| Replay Buffer Capacity | - | - | 2000000 |

**Table 19:** Quadruped Locomotion Hyperparameters

## D DATASET COLLECTION

To collect datasets for each domain and task, we periodically saved the policies at fixed intervals throughout the training process. We then evaluated all the saved policies on 100 episodes for each domain and task. Based on these evaluations, we created a distribution of median returns and assigned an `expertise` level to each policy as follows:

1. `Novice`: The median return lies within one standard deviation below the mean.

2. `Intermediate`: The median return is within one standard deviation above or below the mean.

3. `Expert`: The median return is one standard deviation above the mean or higher.

In some cases, certain domains or tasks were too challenging, resulting in no policies of a given skill level. In such instances, we only provide a `novice` dataset.

## E DATASET INFORMATION

1. Dataset documentation and intended uses:

   - The A2Perf datasets consist of data collected from three simulated environments: computer chip floorplanning, web navigation, and quadruped locomotion. The data was generated by running reinforcement learning policies at various stages of training, capturing the experiences of these policies interacting with the respective environments. The datasets are intended for use in offline reinforcement learning, imitation learning, and hybrid approaches, allowing researchers to evaluate and compare different algorithms without the need for online data collection.

2. Dataset availability:

   - The datasets can be accessed at:
     - Circuit Training: `https://drive.google.com/drive/folders/1UMhLlnYmfbnjBPN_JwVy4YXDUahXrWf6`
     - Quadruped Locomotion: `https://drive.google.com/drive/folders/1n1BJFip-reSPif8Bv3jXAnSOgfQAEje7`
     - Web Navigation: `https://drive.google.com/drive/folders/13EmCscVatl7Q5EFdWFRpwKlA2yRfonE5`

3. Data format and usage:

   - The datasets are provided in the widely-used HDF5 format, a data model and file format designed for efficient storage and retrieval of large datasets. Detailed instructions on how to read and use the data with the Minari framework are provided at: `https://minari.farama.org/`

4. Licensing:

- The A2Perf datasets are released under the MIT License. The authors confirm that they bear all responsibility in case of violation of rights.

5. Maintenance and long-term preservation:

- The datasets are hosted on a Google Cloud Bucket maintained by the Farama Foundation, a non-profit organization dedicated to supporting open-source machine learning projects. This ensures the long-term availability and accessibility of the datasets for the research community.

## F    WEBSITE GENERATION & AGENT INTERACTION

To generate the set of websites $W$, we first assume a target number of websites, denoted as $N_{\text{websites}}$. Following the approach in Gur et al. [23] (shown in Table 4 of the paper), we consider 42 possible primitives that can be added to a web page and introduce two additional primitives: a "new page" primitive and a "stop" primitive, resulting in a total of 44 primitives.

The website generation process begins with an empty web page. We repeatedly sample uniformly from the 44 primitives and add them to the current page. If the "new page" primitive is selected during the sampling process, we start adding primitives to a new linked page. If the "stop" primitive is selected, we conclude the generation of the current website and proceed to generate the next website, if necessary. This process continues until we have generated the desired number of websites, $N_{\text{websites}}$. Each website in the resulting set $W$ consists of one or more web pages, with each page containing a sampled set of primitives.

We define the difficulty of a web page as the probability of a random agent interacting with the correct primitive(s). The difficulty of page $p_i$ is given by $-\log\left(\frac{n_{\text{active}}}{n_{\text{active}} + n_{\text{passive}}}\right)$, where $n_{\text{active}}$ and $n_{\text{passive}}$ denote the number of active and passive primitives on the page, respectively. The difficulty of an entire sequence of web pages is determined by summing the difficulty of all individual pages it contains. Based on these difficulty calculations, we partition the websites into three difficulty levels. The three levels of difficulty correspond to the probability thresholds of 50%, 25%, and 10% for levels 1, 2, and 3, respectively. Users can select a specific difficulty level of web navigation by executing Python commands such as `env = gym.make("WebNavigation-Difficulty-01-v0", num_websites=1)`, where the `num_websites` argument defines the number of websites that are generated for this environment. At each timestep, the agent can interact with an HTML element on the page, such as modifying the text field or clicking on the element, with the objective of entering correct information into forms and clicking "next" or "submit" to advance between web pages.

