# OpenReview forum: "A2Perf: Real-World Autonomous Agents Benchmark"
_ICLR.cc/2025/Conference — ICLR 2025 Conference Withdrawn Submission_

### Official Review · Reviewer_sAYK · 2024-10-31

**Soundness:** 4
**Presentation:** 4
**Contribution:** 4
**Rating:** 8
**Confidence:** 4

**Summary:**

The paper presents a new open-source library to assess RL solutions to real-world problems through a peculiar but necessary lens: resource efficiency, reliability, generalization and task performance.
The paper shows some results across three real-world problems that show a good sim2real transfer capacity and that would potentially benefit from RL.

**Strengths:**

The paper fills a crucial gap in the RL tooling ecosystem and makes a valid point regarding the usefulness of A2Perf.
The method is carefully explained and well thought.
I enjoyed reading it and found the argument compelling.

**Weaknesses:**

When presenting a project such as A2Perf, one must also show where it would be used within the life cycle of a (research or production) project. For instance, some libraries offer tools to train a model and make it clear that the goal is to achieve the best performance given an architecture and an algorithm, others focus on reproducing results in the literature...
Here the goal is very clear but the way the library is to be used (end-to-end vs post-training to validate claims) is a bit unclear.
I believe writing a comprehensive RL library with many SOTA algorithms isn't really the point here, so I'm wondering how one could choose an algorithm, architecture, dataset or problem that is not in the toolkit of the library and use the metrics provided to assess that specific scenario. How easy would that be done compared to training two different algorithms with a common RL library (SB3, torchrl, tianshou)?
It would be good to give the reader some idea of how easy it would be to use A2Perf for a problem that is not already accounted for by the lib.

**Questions:**

There's a typo on L084: it's -> its

I would suggest to swap the order of tables 6 and 7 for clarity.

The energy consumed for the training of a model depends heavily on hardware and software. A compiled model can require significantly less energy than another executed in eager mode (because operations are more efficient and because it is faster to train). What guidance would the authors give for someone assessing the quality of an RL solution, does the library offer the possibility of assessing some metrics before the run is completed (eg, extrapolate the energy cost from the few first epochs) to optimize these metrics before launching a full training loop?

---

### Official Review · Reviewer_bUhm · 2024-11-02

**Soundness:** 2
**Presentation:** 3
**Contribution:** 1
**Rating:** 3
**Confidence:** 4

**Summary:**

The paper proposes a new benchmark termed A2Perf that includes three environments: chip floorplanning, web navigation and quadruped locomotion. The benchmark uses several performance metrics including a newly proposed metric named data cost that measures the cost of collecting a dataset for offline learning. Other metrics such as system performance and reliability ought to test for practical applicability of algorithms. The work provides an evaluation on all three environments using a set of the proposed metrics on each. The work ends with an experimental section concluding that online RL generally has higher energy cost, offline methods have higher data cost. A final experiment demonstrates the pros and cons of RL methods using reliability metrics.

**Strengths:**

1. Clarity
a) Overall the text is well written and easy to follow.
b) The tables are useful in section three provide a nice overview over the proposed metrics and their utility.

2. Motivation
a) The work is motivated by recent advances to produce more reliable measures for success in RL. This motivation is quite timely and this work makes progress toward the challenge of more consistent experimentation.

3. Experiments
a) I think the experimental section highlights several interesting properties of the suggested metrics such as the discrepancy between training and deployment stability of PPO and SAC.

**Weaknesses:**

1. Motivation
a) While I do find the general motivation of this work timely, it is not clear to me why these specific environments were chosen and how they relate.

2. Claims and Novelty
a) It seems that the novelty of this manuscript is limited as most metrics are already known and the environments that are being considered already exist. It seems that the main contribution is to assign existing metrics to existing benchmarks. The exception is the novel data metric that is being introduced. However, as I outline later I’m not convinced of the usefulness of this metric.
b) The motivation for the work seems to be largely based on the claim that evaluation in other benchmarks is missing. One of the initial claims is that “There is currently no benchmarking suite that defines the environments, datasets, and metrics which can be used to develop reference implementations and seed leaderboards with baselines” (L18) This seems to be a very strong statement given the rise of a large number of benchmarks recently.
c) “Most [other benchmarks] only evaluate an agent’s raw performance on the same task on which it was trained, without considering numerous other metrics that matter in real-world production training and deployment scenarios.” (L46) There are various environments and datasets with proposed metrics that measure benchmark related quantities. Some examples:
* [1b, 2b] Continual learning benchmarks measure metrics such as forgetting and collapse.
* [3b] Suggests to evaluate various dataset sizes which seems similar to the data metric proposed in this manuscript.
* [4b, 13b] Measure constraint violation for practical safety features.
d) I am going to interpret Table 1 in the paper as a claim. I do not agree that Safety Gym and CoinRun do not measure reliability. Safety gym explicitly measures violations of constraints which can be interpreted as reliable. The number of tasks an agent can solve out of distribution in CoinRun should have to count as a measure of how reliable a system is to perturbation.  Similarly it is unclear to me why Safety Gym tasks are considered realistic but DM Control is not.

3. Related work
a) The treatment of related work could be improved.
* Given the recent rise of new benchmarks after the introduction of the datasets and benchmarks track at NeurIPS. 8/9 of the benchmarks listed in Table 1 are from 2020 or older. Here is a non-exhaustive list of recent potentially relevant benchmarks [1b, 2b, 3b, 4b, 7b, 8b, 9b, 10b, 11b, 12b, 13b] and datasets [14b, 15b, 16b, 17b]
* The reliability metrics only get cited in section 3 even though the intro states "our reliability metrics" (L92) which might lead to confusion about attribution.

4. Metrics
a) Most of the suggested metrics already exist, see [8, 12, 50] in the paper. Systems performance metrics can quite easily be tracked using frameworks such was Weights&Biases nowadays too.
b) The data cost metric seems inconsistent since any method can use a fixed dataset that already exist. The major upside of using datasets is that the cost of data collection is amortized over time. Once a dataset exists, it can be re-used and it is unclear to me why existing data should be tied to specific methods. Further, it seems odd that all RL agents show 0 data cost as they do need to collect data. In general, RL sample efficiency is significantly lower and it seems incorrect to claim that offline methods have lower data cost. I suppose the argument is supposed to be that this data collection requires human effort to gather expert demonstrations but then the metric is measured using energy consumed to train RL policies which do not require human effort.
c) It seems a little contrived to consider system metrics algorithm dependent rather than hyperparameter dependent. For instance, I could simply change the batch size of my DDQN algorithm an increase the VRAM usage. In section 5.2, if I simply decrease the number of parallel processes, the peak RAM usage would be lower but other metrics would not be affected. I think in order for this to be useful, clearer guidelines are needed on how exactly to measure these metrics.

Overall, I agree with the final assessment that we need more holistic metrics to measure progress in L532. However, it's not entirely clear what unique advantages this benchmark offers beyond existing methods. I think the scientific contributions are limited and the proposed metrics either already exist, are possibly of low utility or at least require more detailed elaboration on how to use them. The benchmark environments also already exist. The experiments provide an interesting study but do not provide sufficient novel insights for me to argue acceptance to ICLR at this time.

[1b] Continual World: A Robotic Benchmark For Continual Reinforcement Learning. Maciej Wolczyk, Michał Zając, Razvan Pascanu, Łukasz Kuciński, Piotr Miłoś. NeurIPS 2021.
[2b] LIBERO: Benchmarking Knowledge Transfer for Lifelong Robot Learning. Bo Liu, Yifeng Zhu, Chongkai Gao, Yihao Feng, Qiang Liu, Yuke Zhu, Peter Stone. NeurIPS 2023.
[3b] Tongzhou Mu, Zhan Ling, Fanbo Xiang, Derek Yang, Xuanlin Li, Xuanlin Li, Stone Tao, Zhiao Huang, Zhiwei Jia, and Hao Su. Maniskill: Generalizable manipulation skill bench-mark with large-scale demonstrations. NeurIPS D&B 2024.
[4b] Alex Ray, Joshua Achiam, and Dario Amodei. Benchmarking Safe Exploration in Deep Reinforcement Learning. 2019.
[5b] Model-Based Reinforcement Learning for Atari. Łukasz Kaiser, Mohammad Babaeizadeh, Piotr Miłos, Błażej Osiński, Roy H Campbell, Konrad Czechowski, Dumitru Erhan, Chelsea Finn, Piotr Kozakowski, Sergey Levine, Afroz Mohiuddin, Ryan Sepassi, George Tucker, Henryk Michalewski. ICLR 2020.
[6b] Evgenii Nikishin, Max Schwarzer, Pierluca D’Oro, Pierre-Luc Bacon, and Aaron Courville. The primacy bias in deep reinforcement learning. ICML 2022.
[7b] Ossama Ahmed, Frederik Träuble, Anirudh Goyal, Alexander Neitz, Manuel Wuthrich, Yoshua Bengio, Bernhard Schölkopf, and Stefan Bauer. CausalWorld: A robotic manipulation benchmark for causal structure and transfer learning. ICLR 2021.
[8b] Jorge A. Mendez, Marcel Hussing, Meghna Gummadi, and Eric Eaton. CompoSuite: A compositional reinforcement learning benchmark. CoLLAs 2022.
[9b] Xavier Puig, Eric Undersander, Andrew Szot, Mikael Dallaire Cote, Tsung-Yen Yang, Ruslan Partsey, Ruta Desai, Alexander Clegg, Michal Hlavac, So Yeon Min, Vladimír Vondruš, Theophile Gervet, Vincent-Pierre Berges, John M Turner, Oleksandr Maksymets, Zsolt Kira, Mrinal Kalakr ishnan, Jitendra Malik, Devendra Singh Chaplot, Unnat Jain, Dhruv Batra, Akshara Rai, and Roozbeh Mottaghi. Habitat 3.0: A co-habitat for humans, avatars, and robots. ICLR 2024.
[10b] DACBench: A Benchmark Library for Dynamic Algorithm Configuration. Theresa Eimer, André Biedenkapp, Maximilian Reimer, Steven Adriaensen, Frank Hutter, Marius Lindauer. ICJAI 2021.
[11b] Clément Bonnet, Daniel Luo, Donal Byrne, Shikha Surana, Sasha Abramowitz, Paul Duckworth, Vincent Coyette, Laurence I. Midgley, Elshadai Tegegn, Tristan Kalloniatis, Omayma Mahjoub, Matthew Macfarlane, Andries P. Smit, Nathan Grinsztajn, Raphael Boige, Cemlyn N. Waters, Mohamed A. Mimouni, Ulrich A. Mbou Sob, Ruan de Kock, Siddarth Singh, Daniel Furelos Blanco, Victor Le, Arnu Pretorius, and Alexandre Laterre. Jumanji: a diverse suite of scalable reinforcement learning environments in jax, 2024.
[12b] Heinrich Küttler, Nantas Nardelli, Alexander H. Miller, Roberta Raileanu, Marco Selvatici, Edward Grefenstette, and Tim Rocktäschel. The NetHack Learning Environment. NeuRIPS 2020.
[13b] Zhaocong Yuan, Adam W. Hall, Siqi Zhou, Lukas Brunke, Melissa Greeff, Jacopo Panerati, and Angela P. Schoellig. Safe-control-gym: A unified benchmark suite for safe learning-based control and reinforcement learning in robotics. IEEE Robotics and Automation 2022.
[14b] Zuxin Liu, Zijian Guo, Haohong Lin, Yihang Yao, Jiacheng Zhu, Zhepeng Cen, Hanjiang Hu, Wenhao Yu, Tingnan Zhang, Jie Tan, et al. Datasets and benchmarks for offline safe reinforcement learning. Journal of Data-centric Machine Learning Research 2024.
[15b] Rong-Jun Qin, Xingyuan Zhang, Songyi Gao, Xiong-Hui Chen, Zewen Li, Weinan Zhang, and Yang Yu. NeoRL: A near real-world benchmark for offline reinforcement learning. NeurIPS 2022 D&B.
[16b] Yun Qu, Boyuan Wang, Jianzhun Shao, Yuhang Jiang, Chen Chen, Zhenbin Ye, Lin Liu, Yang Jun Feng, Lin Lai, Hongyang Qin, Minwen Deng, Juchao Zhuo, Deheng Ye, Qiang Fu, Yang Guang, Yang Wei, Lanxiao Huang, and Xiangyang Ji. Hokoff: Real game dataset from honor of kings and its offline reinforcement learning benchmarks. NeurIPS 2023 D&B.
[17b] Robotic Manipulation Datasets for Offline Compositional Reinforcement Learning. Marcel Hussing, Jorge A. Mendez, Anisha Singrodia, Cassandra Kent, Eric Eaton. RLC 2024.

**Questions:**

Q1 According to your classification, what is the point at which we should consider a task a realistic task?
Q2.What is the main differentiator between this work and existing work that considers realistic scenarios, assuming that I could just use reliability or carbon metric implementations from other frameworks inside those implementations?
Q3. In Table 5, why does generalization not qualify as a metric that has a deviation measure?
Q4. Can you elaborate why standard deviation was chosen as the measure of variance in the experiments?

---

### Official Review · Reviewer_2KBG · 2024-11-03

**Soundness:** 2
**Presentation:** 2
**Contribution:** 2
**Rating:** 5
**Confidence:** 4

**Summary:**

This paper introduces A2Perf, a benchmarking suite for "autonomous agents" with environments in chip floorplanning, web navigation, and quadruped locomotion (simulation). A2Perf includes four metric categories - *1) data cost, application performance (including generalization), 2) system resource, 3) efficiency and 4) reliability* for comparing learning approaches. A2Perf is open-source, extensible, and provides baseline algorithms for fair comparison. The authors attempt to incorporate practical metrics with real-world deployment considerations in mind.

**Strengths:**

1. I appreciate the authors' focus on real-world deployment, as most studies evaluate an agent’s performance solely on its training task, neglecting practical considerations essential for effective training and deployment.
2. Consideration of data cost and energy requirements during training and inference.
3. Proposal of standardized metrics applicable across various task domains
4. Open-source code, which is fairly easy to read and understand

**Weaknesses:**

1. The authors aim to create a unified benchmark focused on real-world application and deployability, with the goal of it being widely adopted by the research community. However, the exact contribution is unclear—they primarily use pre-existing benchmarks and metrics previously proposed/used in other works. Is the novelty here simply the combination of these elements in a single accessible suite? While this can prove valuable, it still feels limited by the small set of available environments.
2. Although the term 'autonomous agent' is used frequently here, it remains undefined. How much human intervention is permitted? Are there separate training and testing phases involving human input? Is a continual learning setup, as described in [1], considered?
3. Generalization as a key metric—This metric seems vaguely defined. Why should it be reasonable to expect a policy trained on task 1 to perform well on task 2 without additional training or fine-tuning? Does generalization here only refer to task performance, or does it also encompass task representation and knowledge of the environment?
4. Although the paper addresses real-world applications, its evaluation is limited to simulation benchmarks. For instance, how would evaluation differ with a real robot dog? What about factors like reset time between episodes or the need for human intervention?
5. The paper also overlooks the robustness of hyper-parameters. Given that deep RL algorithms are notoriously sensitive to hyperparameter choices, shouldn't this be considered part of the training costs?
6. Justification for metrics specified in Table 3 is insufficient. The authors should consider additional metrics from [3] and [4] when evaluating application performance.
7. The paper lacks citations for large-scale benchmarks such as RoboHive [5], which supports a wide range of robot embodiments.

**References**
1. Dohare, S., Hernandez-Garcia, J. F., Lan, Q., Rahman, P., Mahmood, A. R., & Sutton, R. S. (2024). Loss of plasticity in deep continual learning. Nature, 632(8026), 768-774.
2. Ceron, J. S. O., Araújo, J. G. M., Courville, A., & Castro, P. S. (2024). On the consistency of hyper-parameter selection in value-based deep reinforcement learning. Reinforcement Learning Journal, 3, 1037–1059.
3. Agarwal, R., Schwarzer, M., Castro, P. S., Courville, A. C., & Bellemare, M. (2021). Deep reinforcement learning at the edge of the statistical precipice. Advances in neural information processing systems, 34, 29304-29320.
4. Patterson, A., Neumann, S., White, M., & White, A. (2023). Empirical design in reinforcement learning. arXiv preprint arXiv:2304.01315.
5. Kumar, V., Shah, R., Zhou, G., Moens, V., Caggiano, V., Gupta, A., & Rajeswaran, A. (2024). Robohive: A unified framework for robot learning. Advances in Neural Information Processing Systems, 36.

**Questions:**

1. Is Table 7 a subset of Table 5? Am I missing any differences between them?
2. Do the authors have any suggestions for metrics to assess the safety of the learned behavior? Note that I'm not expecting this to be addressed in the current paper
3.  What is $\alpha$ in $CVaR_\alpha$? $\alpha$ is not used in the definition in Table 3.
4. Why is the number of CPU cores not considered? Many implementations depend on CPU parallelism as well.
5. In real-world deployments, powerful computing resources are often available for training (e.g., clusters), while only resource-constrained devices are available during deployment (e.g., edge devices on mobile robots). This discrepancy is not currently reflected in the proposed metrics. Do the authors have any thoughts on this matter?
6. Similar to the previous question, how would Sim2Real approaches fit into this benchmark?
7.  Why not integrate with existing benchmarks like this instead of focusing on a specific implementation, such as quadruped locomotion?

---

### Official Review · Reviewer_DcKw · 2024-11-08

**Soundness:** 2
**Presentation:** 2
**Contribution:** 2
**Rating:** 3
**Confidence:** 4

**Summary:**

The paper proposes an open-source suite of three RL benchmark environments purported to be a more realistic model of real-world tasks than prior RL benchmarks. Metrics that capture success, efficiency, and reliability are proposed for RL methods. DDQN, PPO, and SAC are evaluated, as well as BC on "expert" data drawn from trained checkpoints.

**Strengths:**

- Environment domains seem more realistic than many current benchmarks used in the community
- Metrics besides task performance are presented, such as reliability and efficiency

**Weaknesses:**

- Limited baselines were evaluated (one offline, BC, and two online baselines per environment).
- The authors did include standard deviations, but it is unclear which results are statistically significant. The paper would benefit from stating this explicitly.
- The paper would benefit from summaries across the environments about which methods perform the best (summary figure, bolding, etc.)
- Given this paper is proposing a benchmark without any algorithmic advances, the limited evaluation and issues with rigor (see questions) make it difficult to evaluate the contribution

**Questions:**

- Why are different benchmarks evaluated on the environments? All tables show PPO and BC, while only some have SAC and DDQN.
- Likewise, why were different expert dataset qualities (novice, intermediate, etc.) used across the different evaluation tables? How was this determined?
- Appendix D describes how expert data for BC is generated by thresholding the value of rollouts from policy checkpoints. How were these checkpoints selected?
- Why are some of the reliability metrics marked as N/A for BC?

---

### Note · Authors · 2024-11-15

I have read and agree with the venue's withdrawal policy on behalf of myself and my co-authors.